# Spatial cell firing during virtual navigation of open arenas by head-restrained mice

Guifen Chen[1,2†], John Andrew King[3†], Yi Lu[1,2†], Francesca Cacucci[2]*,
Neil Burgess[1,4]*

[1]UCL Institute of Cognitive Neuroscience, University College London, London, United Kingdom; [2]Department of Neuroscience Physiology and Pharmacology, University College London, London, United Kingdom; [3]Department of Clinical Educational Health Psychology, University College London, London, United Kingdom; [4]UCL Institute of Neurology, University College London, London, United Kingdom

**Abstract** We present a mouse virtual reality (VR) system which restrains head-movements to horizontal rotations, compatible with multi-photon imaging. This system allows expression of the spatial navigation and neuronal firing patterns characteristic of real open arenas (R). Comparing VR to R: place and grid, but not head-direction, cell firing had broader spatial tuning; place, but not grid, cell firing was more directional; theta frequency increased less with running speed, whereas increases in firing rates with running speed and place and grid cells' theta phase precession were similar. These results suggest that the omni-directional place cell firing in R may require local-cues unavailable in VR, and that the scale of grid and place cell firing patterns, and theta frequency, reflect translational motion inferred from both virtual (visual and proprioceptive) and real (vestibular translation and extra-maze) cues. By contrast, firing rates and theta phase precession appear to reflect visual and proprioceptive cues alone.
DOI: https://doi.org/10.7554/eLife.34789.001

*For correspondence:
f.cacucci@ucl.ac.uk (FC);
n.burgess@ucl.ac.uk (NB)

†These authors contributed equally to this work

## Introduction

Virtual reality (VR) offers a powerful tool for investigating spatial cognition, allowing experimental control and environmental manipulations that are impossible in the real world. For example, uncontrolled real-world cues cannot contribute to determining location within the virtual environment, while the relative influences of motoric movement signals and visual environmental signals can be assessed by decoupling one from the other (*Tcheang et al., 2011*; *Chen et al., 2013*). In addition, the ability to study (virtual) spatial navigation in head-fixed mice allows the use of intracellular recording and two photon microscopy (*Dombeck et al., 2010*; *Harvey et al., 2009*; *Royer et al., 2012*; *Domnisoru et al., 2013*; *Schmidt-Hieber and Häusser, 2013*; *Heys et al., 2014*; *Low et al., 2014*; *Villette et al., 2015*; *Danielson et al., 2016*; *Cohen et al., 2017*). However, the utility of these approaches depends on the extent to which the neural processes in question can be instantiated within the virtual reality (for a recent example of this debate see *Minderer et al., [2016]*).

The modulation of firing of place cells or grid cells along a single dimension, such as distance travelled along a specific trajectory or path, can be observed as virtual environments are explored by head-fixed mice (*Chen et al., 2013*; *Dombeck et al., 2010*; *Harvey et al., 2009*; *Domnisoru et al., 2013*; *Schmidt-Hieber and Häusser, 2013*; *Heys et al., 2014*; *Low et al., 2014*; *Cohen et al., 2017*) or body-fixed rats (*Ravassard et al., 2013*; *Acharya et al., 2016*; *Aghajan et al., 2015*). However, the two-dimensional firing patterns of place, grid and head-direction cells in real-world open arenas are not seen in these systems, in which the animal cannot physically rotate through 360°.

By contrast, the two-dimensional (2-d) spatial firing patterns of place, head direction, grid and border cells have been observed in VR systems in which rats can physically rotate through 360° (*Aronov and Tank, 2014*; *Hölscher et al., 2005*). Minor differences with free exploration remain, for example the frequency of the movement-related theta rhythm is reduced (*Aronov and Tank, 2014*), perhaps due to the absence of translational vestibular acceleration signals (*Ravassard et al., 2013*; *Russell et al., 2006*). However, the coding of 2-d space by neuronal firing can clearly be studied. These VR systems constrain a rat to run on top of an air-suspended Styrofoam ball, wearing a 'jacket' attached to a jointed arm on a pivot. This allows the rat to run in any direction, its head is free to look around while its body is maintained over the centre of the ball.

These 2-d VR systems retain a disadvantage of the real-world freely moving paradigm in that the head movement precludes use with multi-photon microscopy. In addition, some training is required for rodents to tolerate wearing a jacket. Here, we present a VR system for mice in which a chronically implanted head-plate enables use of a holder that constrains head movements to rotations in the horizontal plane while the animal runs on a Styrofoam ball. Screens and projectors project a virtual environment in all horizontal directions around the mouse, and onto the floor below it, from a viewpoint that moves with the rotation of the ball, following *Aronov and Tank (2014)* and *Hölscher et al. (2005)* (see *Figure 1* and Materials and methods).

We demonstrate that mice can navigate to an unmarked location within an open virtual arena, in a task like a continuous Morris Water Maze task (*Morris et al., 1982*), combining reference memory for an unmarked location with foraging designed to optimize environmental coverage for recording spatial firing patterns. That is, the mice can perceive and remember locations defined by the virtual space. We also show that the system allows expression of the characteristic 2-d firing patterns of place cells, head-direction cells and grid cells in electrophysiological recordings, making their underlying mechanisms accessible to investigation by manipulations of the VR.

## Results

### Navigation in VR

Eleven mice were trained in the virtual reality system (see *Figure 1* and *Figure 1—figure supplement 1*, and Materials and methods). All training trials in the VR and the real square environments from the 11 mice were included in the behavioral analyses below. The mice displayed an initially lower running speed when first experiencing the real-world recording environment (a $60 \times 60$ cm square), but reached a higher average speed after 20 or so training trials. The increase in running speed with experience was similar in the virtual environments (*Figure 1F–H*). Running speeds did not differ between the 60 cm and 90 cm virtual environments used for recording in seven and four of the mice, respectively ($12.01 \pm 2.77$ in 60 cm VR, $14.33 \pm 4.19$ cm/s in 90 cm VR, p=0.29). Running directions in the VR environment showed a marginally greater unimodal bias compared to the real environment (R; *Figure 1K*). Mice displayed a greater tendency to run parallel to the four walls in VR, a tendency which reduced with experience (*Figure 1—figure supplement 2*). They also took straighter, less tortuous, paths in VR than in R, as would be expected from their head-fixation (*Figure 1L–N*).

In the fading beacon task, performance steadily improved across 2–3 weeks of training (*Figure 2D*, one trial per day). They learned to approach the fixed reward location and could do so even after it became completely unmarked (fully faded, see *Figure 2* and *Video 1* of a mouse performing the task, and Materials and methods for details of the training regime).

### Electrophysiology

We recorded a total of 231 CA1 cells from seven mice: 179 cells were classified as place cells in the real environment, 185 cells in the virtual environment, and 154 cells were classified as place cells in both real and virtual environments (see Materials and methods).

We recorded 141 cells in dorsomedial Entorhinal Cortex (dmEC) from eight mice, 82 of them were classified as grid cells in the real environment, 65 of them grid cells in the virtual environment, and 61 were classified as grid cells in both real and virtual environments. Among these 141 recorded cells, 16 cells were quantified as head-direction cells (HDCs) in R, 20 cells were HDCs in VR, with 12 cells classified as HDCs in both real ('R') and virtual reality ('VR') environments. All cells were

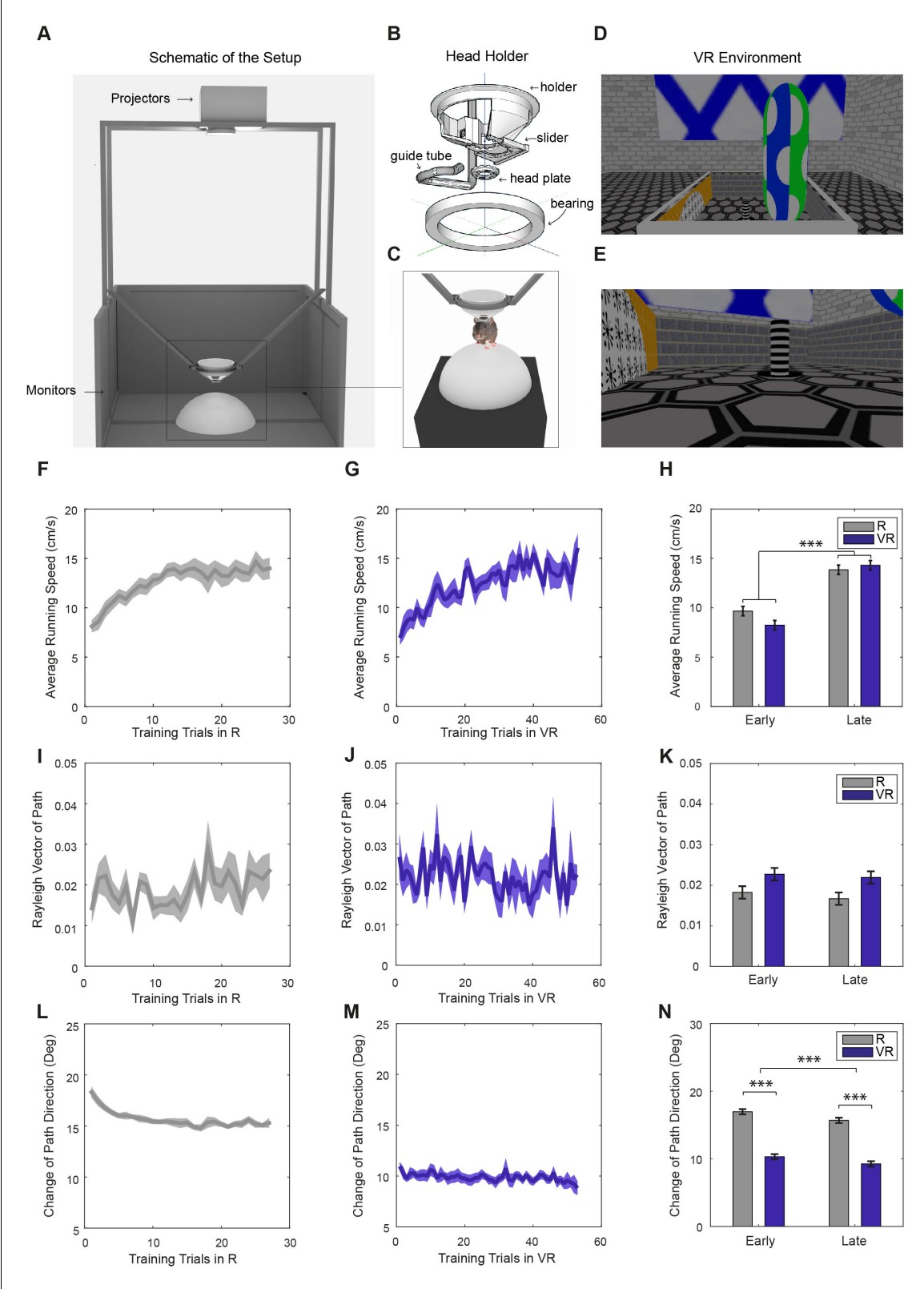

**Figure 1.** Virtual reality setup and behavior within it. (**A**) Schematic of the VR setup (VR square). (**B**) A rotating head-holder. (**C**) A mouse attached to the head-holder. (**D–E**) Side views of the VR environment. (**F–G**) Average running speeds of all trained mice (n = 11) across training trials in real ('R'; **F**) and virtual reality ('VR'; **G**) environments in the main experiment. (**H**) Comparisons of the average running speeds between the first five trials and the last five trials in both VR and R environments, showing a significant increase in both (n = 11, p<0.001, F(1,10)=40.11). (**I–J**) Average Rayleigh vector lengths of

*Figure 1 continued on next page*

*Figure 1 continued*

running direction across training trials in R (I) and VR (J). (K) Comparisons of the average Rayleigh vector lengths of running direction between the first five trials and the last five trials in both VR and R. Directionality was marginally higher in VR than in R (n = 11, p=0.053, F(1,10)=4.82) and did not change significantly with experience. (L–M) Average changes of running direction (absolute difference in direction between position samples) across training trials in R (L) and VR (M). (N) Comparisons of the changes of running direction between the first five and last five trials in both R and VR. Animals took straighter paths in VR than R (n = 11, p<0.001, F(1,10)=300.93), and paths became straighter with experience (n = 11, p<0.001, F(1,10)=26.82). Positions were sampled at 2.5 Hz with 400 ms boxcar smoothing in (I–N). All error bars show s.e.m.

DOI: https://doi.org/10.7554/eLife.34789.002

The following figure supplements are available for figure 1:

**Figure supplement 1.** Example paths in the three training stages and the recording stage.
DOI: https://doi.org/10.7554/eLife.34789.003

**Figure supplement 2.** Directional polar plots of running directions in the VR square environment (n = 11 mice).
DOI: https://doi.org/10.7554/eLife.34789.004

**Figure supplement 3.** Nissl-stained brain sections from the 11 mice in the main experiment.
DOI: https://doi.org/10.7554/eLife.34789.005

recorded while animals randomly foraged in both R and VR environments (see Materials and methods).

Place cells recorded from CA1 showed spatially localized firing in the virtual environment, with similar firing rates in the virtual and real square environments. Place cells had larger firing fields in VR than in R, by a factor 1.44 (field size in VR/field size in R). The spatial information content of firing fields in VR was lower than in R. In addition, the firing of place cells was more strongly directionally modulated in VR than in R (see *Figure 3*). Similar results were seen irrespective of whether recordings took place in the 60 × 60 cm or 90 × 90 cm VR environments (e.g. the place field expansion factor being 1.44 in 90 cm, 1.43 in 60 cm, p=0.66, see *Figure 4—figure supplement 1*).

One possible contribution to apparent directionality in place cell firing could be inhomogeneous sampling of direction within the (locational) firing field. This can be controlled for by explicitly estimating the parameters of a joint place and direction ('pxd') model from the firing rate distribution (*Burgess et al., 2005*). However, using this procedure did not ameliorate the directionality in firing (see *Figure 3*). Further analyses showed that firing directionality increased near to the boundaries in both virtual and real environments (where sampling of direction is particularly inhomogeneous), but that the additional directionality in VR compared to R was apparent also away from the boundaries (see *Figure 3—figure supplement 1*). We investigated further whether the increased directionality of place cell firing in VR was specific to the square VR environment, by performing additional recordings of both place and grid cells while animals foraged in (visually similar) cylindrical and square R and VR environments (in four mice, three new to the experiment, yielding a total of 90 place and nine grid cells). The increased directionality of place cells but not grid cells in VR was present in both cylinder and square environments, supporting the generality of the result (see *Figure 3—figure supplement 2*).

Grid cells recorded in dmEC, showed similar grid-like firing patterns in VR as in R, with similar firing rates and 'gridness' scores. The spatial scale of the grids was larger in VR than in R, with an average increase of 1.42 (grid scale in VR/grid scale in R, n = 6 mice). The spatial information content of grid cell firing was lower in VR than R, as with the place cells. Unlike the place cells, the grid cells showed a slight decrease in directionality from R to VR, although this appears to reflect inhomogeneous sampling of directions within firing fields, as the effect was not seen when controlling for this in a joint 'pxd' model (see *Figure 4*). Similar results were seen irrespective of whether recordings took place in the 60 × 60 cm or 90 × 90 cm VR environments (e.g. the grid scale expansion factor being 1.43 in 60 cm, 1.36 in 90 cm, p=0.78), although there were minor differences (the reduction in spatial information only reaching significance in the 60 × 60 cm VR and that in directional information only reaching significance in the 90 × 90 cm VR, see *Figure 4—figure supplement 1*).

It is possible that low directional modulation of the firing of a grid cell could reflect directionally modulated firing fields with different directional tuning. Accordingly, we checked the directional information in the firing of each field, without finding any difference between R and VR (*Figure 4H*).

To check whether any differences between R and VR could reflect the trial order (VR before R), we recorded additional data from place and grid cells in R and VR on days in which R trials both

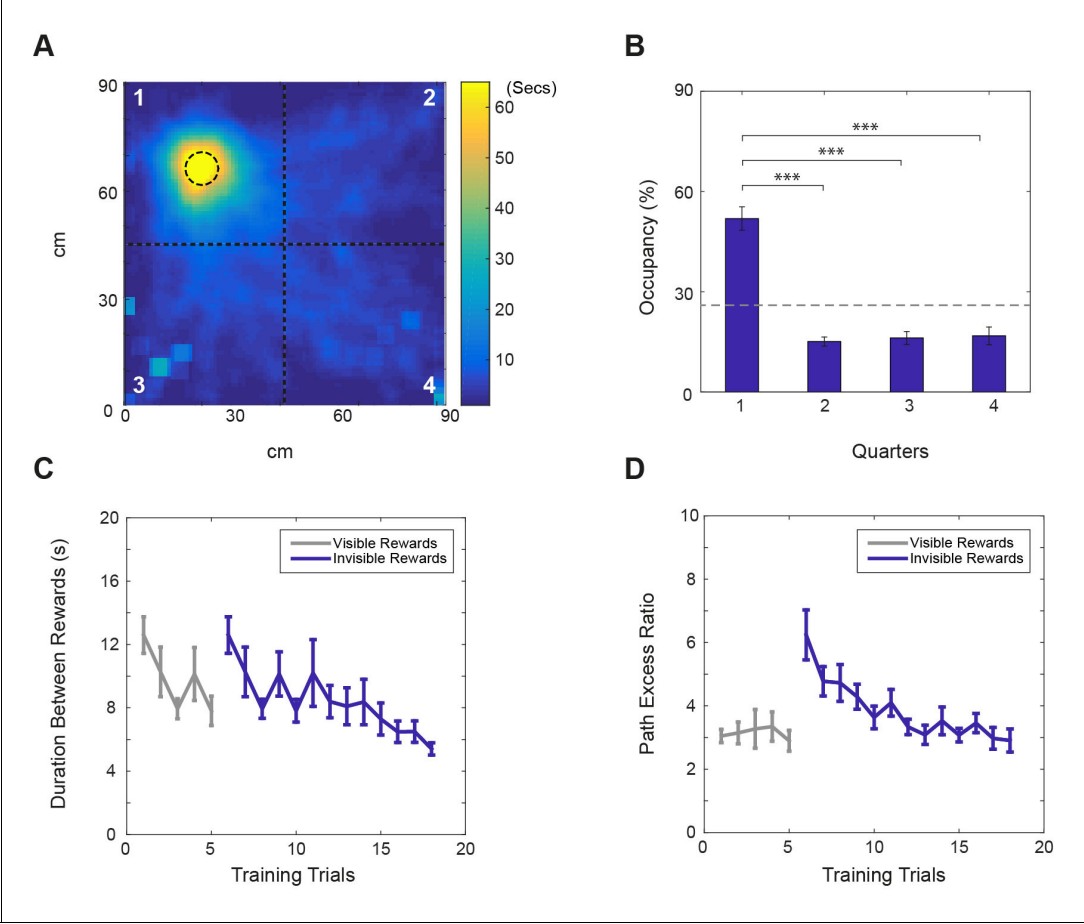

**Figure 2.** Performance on the 'fading beacon' task. (**A**) An example heat map showing the distribution of locations between the third and the fourth rewards during a 40 min trial (mouse#987, trial#24). The dotted circle in the first quadrant shows the location of the faded reward. (**B**) Average time spent (as % of total time) in each quadrant of the VR square (numbered in A) showed a clear bias (n = 11, p<0.001, F(3,30)=39.03), with time spent in the first quadrant was significantly higher than in the others (*** denotes significance at p<0.001, ** at p<0.01). (**C**) Average durations between the third and the fourth rewards across training trials. (**D**) Average path excess ratios between the third and the fourth rewards across training trials (means ± s.e. m). Note that in each set of four rewards, the first, second and third rewards appeared at random locations in the virtual square, marked by visual beacons, the fourth reward was located at a fixed location. Grey lines show trials when the fixed-location rewards were marked by visual beacons. Blue lines show trials when the fixed rewards were not marked. See Supplementary video.

DOI: https://doi.org/10.7554/eLife.34789.006

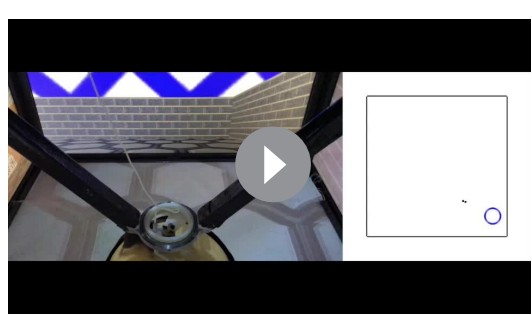

**Video 1.** example of a mouse performing the 'fading beacon' task.
DOI: https://doi.org/10.7554/eLife.34789.007

preceded and followed VR trials (in four mice, three new to the experiment). We also included analysis of the first 20 mins of VR trials (matching the length of R trials, see *Figure 4—figure supplement 2*). Under these conditions, the differences in firing properties between R and VR are similar to those shown in *Figures 3* and *4*, again indicating generality. However, the 20 grid cells in this group did show lower gridness scores in VR than R, and 43 cells were classified as grid cells in R but only 24 as grid cells in VR. Thus, grid cell firing patterns can be sensitive to the use of VR and the inherent conflict between virtual and uncontrolled cues to translation. The extra sensitivity in the second group of animals

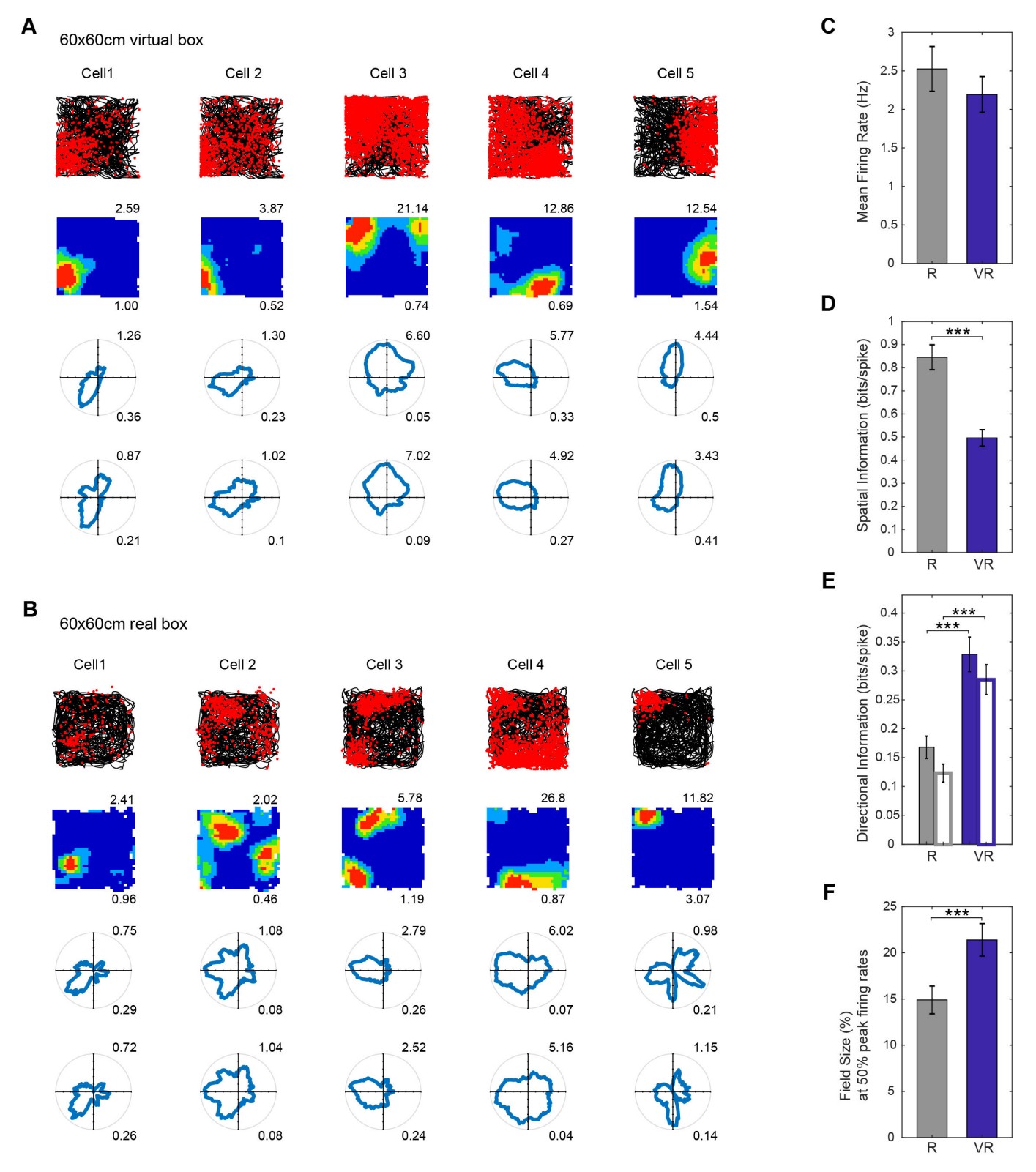

**Figure 3.** Place cell firing in real and virtual environments. (A–B) The same five place cells recorded in a 60 × 60 cm virtual square (A) and in a 60 × 60 cm real square (B, one cell per column). Top row: 40 min running trajectory (black line) with red dots showing the locations of spikes; 2nd row, firing rate maps, maximum firing rate (Hz) shown at top right, spatial information (bits/spike) bottom right; third and fourth row: polar plots of directional firing rates (third row: standard binning; fourth row: after fitting a joint 'pxd' model to account for in homogeneous sampling), maximum firing rate top

*Figure 3 continued on next page*

*Figure 3 continued*

right, directional information bottom right. (C–F) Comparison between R (grey bars) and VR (blue bars): (C) Mean firing rates, higher in R than VR but not significantly so (n = 154, t(153)=1.67, p=0.10); (D) Spatial information, significantly higher in R than in VR (n = 154, t(153)=8.90, p<0.001); (E) Directional information rates using standard (solid bars) and pxd binning (open bars), greater in VR than R (standard n = 154, t(153)=6.45, p<0.001; pxd, n = 154, t(153)=7.61, p<0.001). (F) Field sizes (bins with firing above 50% of peak firing rate, as a proportion to the size of the test environment), were larger in VR than in R (n = 154, t(153)=4.38, p<0.001).

DOI: https://doi.org/10.7554/eLife.34789.008

The following figure supplements are available for figure 3:

**Figure supplement 1.** Directional information of place cell firing (bits/spike) as a function of the distance from the nearest wall (as % of the width of environment) in real (A) and virtual (B) environments (154 place cells from 11 animals).

DOI: https://doi.org/10.7554/eLife.34789.009

**Figure supplement 2.** VR and R trials in square and cylindrical environments.

DOI: https://doi.org/10.7554/eLife.34789.010

might reflect their greater age at test (mice with grid cells, main experiment: n = 8, age = 25.4 ± 4.3 weeks; additional experiment: n = 3, age = 40.1 ± 11.2 weeks; t(9)=-3.34, p<0.01) but this would require further verification.

We also recorded head-direction cells in the dmEC, as previously reported in rats (*Sargolini et al., 2006*) and mice (*Fyhn et al., 2008*). These cells showed similar firing rates in VR and R, with similar tuning widths (see *Figure 5*). The relative differences in the tuning directions of simultaneously recorded directional cells was maintained between R and VR, even though the absolute tuning direction was not (see *Figure 5—figure supplement 1*).

The translational movement defining location within the virtual environment purely reflects feedback (visual, motoric and proprioceptive) from the virtual reality system, as location within the real world does not change. However, the animal's sense of orientation might reflect both virtual and real-world inputs, as the animal rotates in both the real and virtual world. To check for the primacy of the controlled virtual inputs versus potentially uncontrolled real-world inputs (e.g. auditory or olfactory), we performed a 180° rotation of the virtual environment and the mouse's entry to it between trials. Note that the geometry of the apparatus itself (square configuration of screens, overhead projectors on either side) would conflict with rotations other than 180°. In separate trials, we observed a corresponding rotation of the virtual firing patterns of place, grid and head-direction cells, indicating the primacy of the virtual environment over non-controlled real world cues (see *Figure 6*). While all the grid and head direction cells followed the rotation of VR cues (and entry point), a small percentage of place cells (7/141; 5%) did not. These place cells show much lower spatial information scores in both the R and VR conditions (see *Figure 6—figure supplement 1*), indicating that their lack of rotation might be the result of their weaker or less stable spatial tuning to the proximal environmental cues that were rotated.

The animal's running speed is known to correlate with the firing rates of cells, including place cells, grid cells and (by definition) speed cells (*Sargolini et al., 2006*; *McNaughton et al., 1983*; *Kropff et al., 2015*), and with the frequency of the local field potential theta rhythm (*McFarland et al., 1975*; *Rivas et al., 1996*; *Sławińska and Kasicki, 1998*). So these experimental measures can give us an independent insight into perceived running speed. We found that the slope of the relationship between theta frequency and running speed was reduced within the VR compared to R, while this was not the case for the firing rates of place, grid and speed cells (see *Figure 7*). However, the changes in grid scale and theta frequency in virtual versus real environments did not correlate with each other significantly across animals. There was an effect of running speed on the sizes of place and grid fields that was similar in R and VR, but was not the monotonic relationship that would be predicted by an effect of (speed-related) theta frequency on field size (see *Figure 7—figure supplement 1*).

Finally, an important aspect of place and grid cell firing is the temporal coding seen in the theta phase precession of firing in 1-d (*O'Keefe and Recce, 1993*; *Hafting et al., 2008*) and 2-d (*Climer et al., 2015*; *Jeewajee et al., 2014*) environments. We calculated the theta-band frequency modulation of firing of theta-modulated cells (see Materials and methods for details) and compared it to the LFP theta frequency in R and VR. This analysis shows that, despite the lower overall frequencies in VR, theta modulated firing frequency is slightly higher than the LFP frequency in both R and

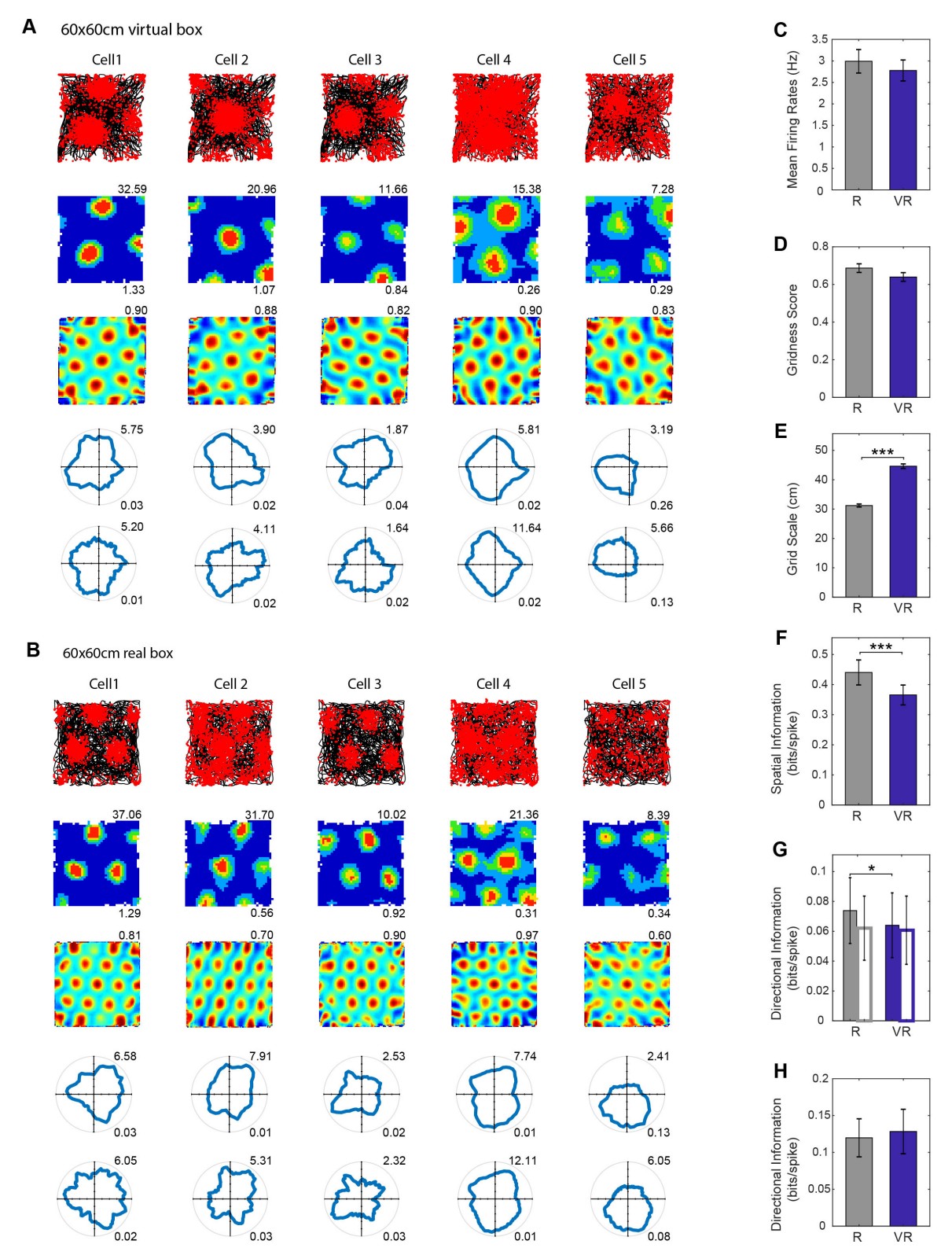

**Figure 4.** Grid cell firing in real and virtual environments. (A–B) The same five grid cells simultaneously recorded in a 60 × 60 cm virtual square (A) and in a 60 × 60 cm real square (B, one cell per column). Top row: 40 min running trajectory (black line) with red dots showing the locations of spikes; second row, firing rate maps, maximum firing rate (Hz) shown top right, spatial information (bits/spike) bottom right; third row: spatial autocorrelations, gridness scores top right; fourth and fifth rows: polar plots of directional firing rates (fourth row: standard binning; fifth row: 'pxd' binning to account for

*Figure 4 continued on next page*

*Figure 4 continued*

inhomogeneous sampling), maximum firing rate top right, directional information bottom right. (C–H) Comparison between R (grey bars) and VR (blue bars): (C) Mean firing rates, higher in R than VR but not significantly so (n = 61, t(60)=1.71, p=0.09); (D) Gridness scores, higher in R than VR but not significantly so (n = 61, t(60)=1.67, p=0.10); (E) Grid scales, larger in VR than in R (n = 61, t(60)=15.52, p<0.001); (F) Spatial information in bits/spike, higher in R than VR (n = 61, t(60)=4.12, p<0.001); (G) Directional information. Grid cell firing was slightly more directional in VR than in R (n = 61, t(60) =2.04, p<0.05), but the difference disappeared when calculated using pxd plots (open bars, n = 61, t(60)=0.32, p=0.75); (H) Directional information in individual grid firing fields, not significant difference between the R and VR trials based on pxd plots (n = 61, t(60)=0.53, p=0.60).

DOI: https://doi.org/10.7554/eLife.34789.011

The following figure supplements are available for figure 4:

**Figure supplement 1.** Breakdown of spatial firing properties in 60 × 60 cm or 90 × 90 cm VR environments.

DOI: https://doi.org/10.7554/eLife.34789.012

**Figure supplement 2.** Trial order and trial length effects on comparing firing properties across VR and R environments in additional data from four mice.

DOI: https://doi.org/10.7554/eLife.34789.013

VR, consistent with theta phase precession (see *Figure 8A*). In addition, we directly analysed phase precession in place and grid cells with clear theta-modulation, finding normal 2-d phase precession in both VR and in R (as in 1-d VR [*Harvey et al., 2009*]). The phase precession slope is lower in VR, consistent with larger firing fields overall in VR environments, but correlations between slope and field size only reached significance for place cells in R (n = 38, r = 0.46, p<0.01; all other p>0.3, see *Figure 8B–E*). These results indicate that theta phase precession in place and grid cells is independent of linear vestibular acceleration signals and the absolute value of theta frequency.

## Discussion

We have demonstrated the ability of a novel mouse virtual reality (VR) system to allow expression of spatial learning and memory in open environments, following related work in rats (*Aronov and Tank, 2014*; *Hölscher et al., 2005*; *Cushman et al., 2013*). Specifically, in the fading-beacon task, mice learned to return to a rewarded spatial location from multiple novel starting locations which (once the beacon had faded) was not marked by any proximal sensory stimulus, that is requiring allocentric spatial memory.

Importantly, we also demonstrated that the VR system allows expression of the characteristic spatially modulated firing patterns of place, grid and head-direction cells in open arenas. Thus, it passes the first pre-requisite as a tool for studying the mechanisms behind the two dimensional firing patterns of these spatial cells, following previous systems for rats that also allow physical rotation of the animal (*Aronov and Tank, 2014*; *Hölscher et al., 2005*). Head-fixed or body-fixed VR systems have been very successful for investigating the one-dimensional spatial firing patterns of place cells (*Chen et al., 2013*; *Dombeck et al., 2010*; *Harvey et al., 2009*; *Cohen et al., 2017*; *Ravassard et al., 2013*; *Acharya et al., 2016*; *Aghajan et al., 2015*) or grid cells (*Domnisoru et al., 2013*; *Schmidt-Hieber and Häusser, 2013*; *Heys et al., 2014*; *Low et al., 2014*), for example, modulation of firing rate by the distance along a linear trajectory. But the two-dimensional firing patterns of place, grid or head direction cells are not seen in these systems.

Although the characteristic firing patterns of spatial cells were expressed within our VR system, there were also some potentially instructive differences in their more detailed properties between VR and a similar real environment (R), which we discuss below.

The spatial scale of the firing patterns of both place cells and grid cells was approximately 1.4 times larger in VR compared to the real environment ('R', *Figures 3* and *4*; see also *Aronov and Tank, [2014]*). Along with the increased scale of place and grid cell responses in VR, there was a reduction in the dependence of theta frequency on running speed. The LFP theta frequency reflects a contribution from vestibular translational acceleration signals (*Ravassard et al., 2013*; *Russell et al., 2006*) which will be absent in our VR system. However, there was no change in the increasing firing rates of place, grid and speed cells with running speed in VR (*Figure 7*), indicating an independence from vestibular translational acceleration cues. Thus, it is possible that the absence of linear acceleration signals affects both LFP theta rhythmicity and the spatial scale of firing patterns, but there was no evidence that the two were directly related.

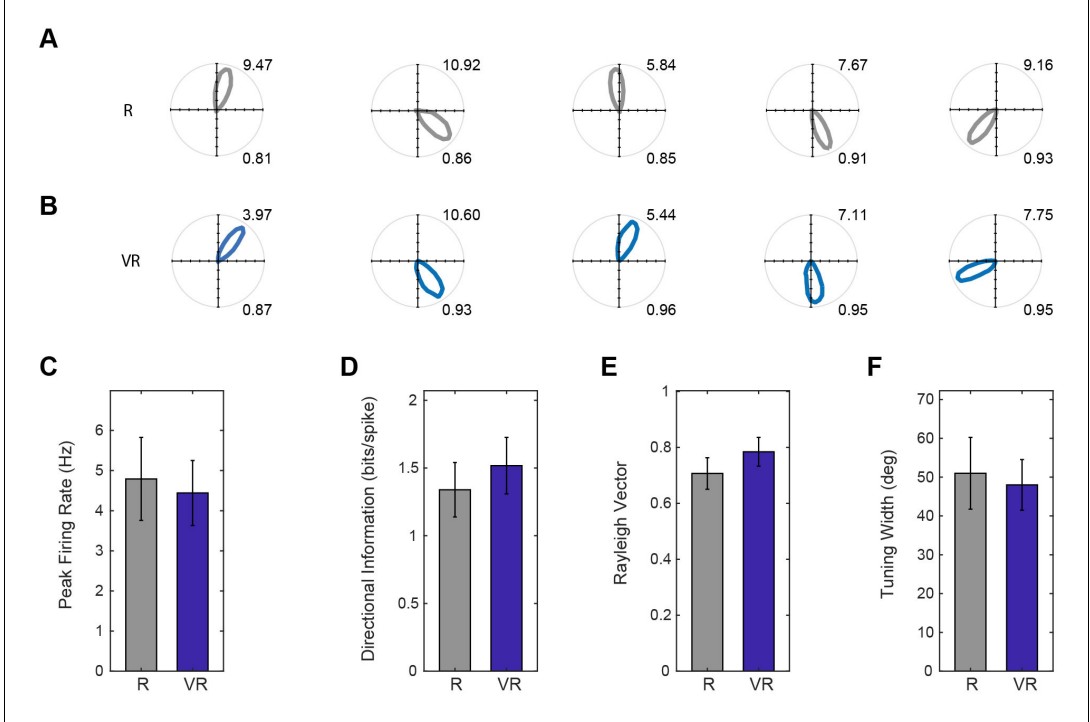

**Figure 5.** Head direction cell firing in real and virtual environments. (A–B) Polar plots of the same five HD cells in dmEC simultaneously recorded in R (A) and VR (B, one cell per column). Maximum firing rates are shown top right, Rayleigh vector length bottom right. (C–F) Comparisons of basic properties of HD cells in dmEC between R and VR. There were no significant differences in peak firing rates (t(11)=0.65, p=0.53; (C); directional information (t(11)=1.38, p=0.19; D); Rayleigh vector length (t(11)=1.69, p=0.12; E); and tuning width (t(11)=0.48, p=0.64; F).
DOI: https://doi.org/10.7554/eLife.34789.014

The following figure supplement is available for figure 5:

**Figure supplement 1.** Eleven directional cells recorded in dmEC.
DOI: https://doi.org/10.7554/eLife.34789.015

Finally, uncontrolled distal cues, such as sounds and smells, and the visual appearance of the apparatus aside from the screens (the edge of the ball, the edges of the screens) will conflict with virtual cues indicating self-motion. Thus, increased firing field size could also reflect broader tuning or reduced precision due to absent or conflicting inputs, consistent with the reduced spatial information seen in place and grid cell firing patterns (*Figures 3* and *4*), and potentially a response to spatial uncertainty (*Towse et al., 2014*). Nonetheless, more grid cells met the spatial firing criteria in R than in VR, which was not true for place or head-direction cells, and lower gridness scores were observed in VR in a group of four older mice (*Figure 4—figure supplement 2*). This would be consistent with the presence of conflict in translational cues in VR (e.g. vestibular versus optic flow) if grid cell firing specifically reflects translational self-motion (*McNaughton et al., 2006*).

The head direction cells do not show broader tuning in the VR (*Figure 5*), probably because there is no absence of vestibular rotation cues and no conflict with distal real-world cues, as the mice rotate similarly in the virtual and real world. We note however, that spatial firing patterns follow the virtual cues when the virtual cues and entry point are put into conflict with uncontrolled real-world cues (*Figure 6*).

Place cell firing in VR showed an increased directionality compared to the real environment. One possible explanation, that the apparent directionality reflected inhomogeneous sampling of directions in the firing field, was not supported by further analyses (*Figure 3E*). Another possible explanation, that differences in directionality reflected the specific features of the VR square, was not supported by further experiments in a second virtual environment (see *Figure 3—figure supplement 2*). A potential benefit of VR is an absence of local sensory cues to location. Experimenters typically work hard to remove consistent uncontrolled cues from real-world experiments (e.g. cleaning

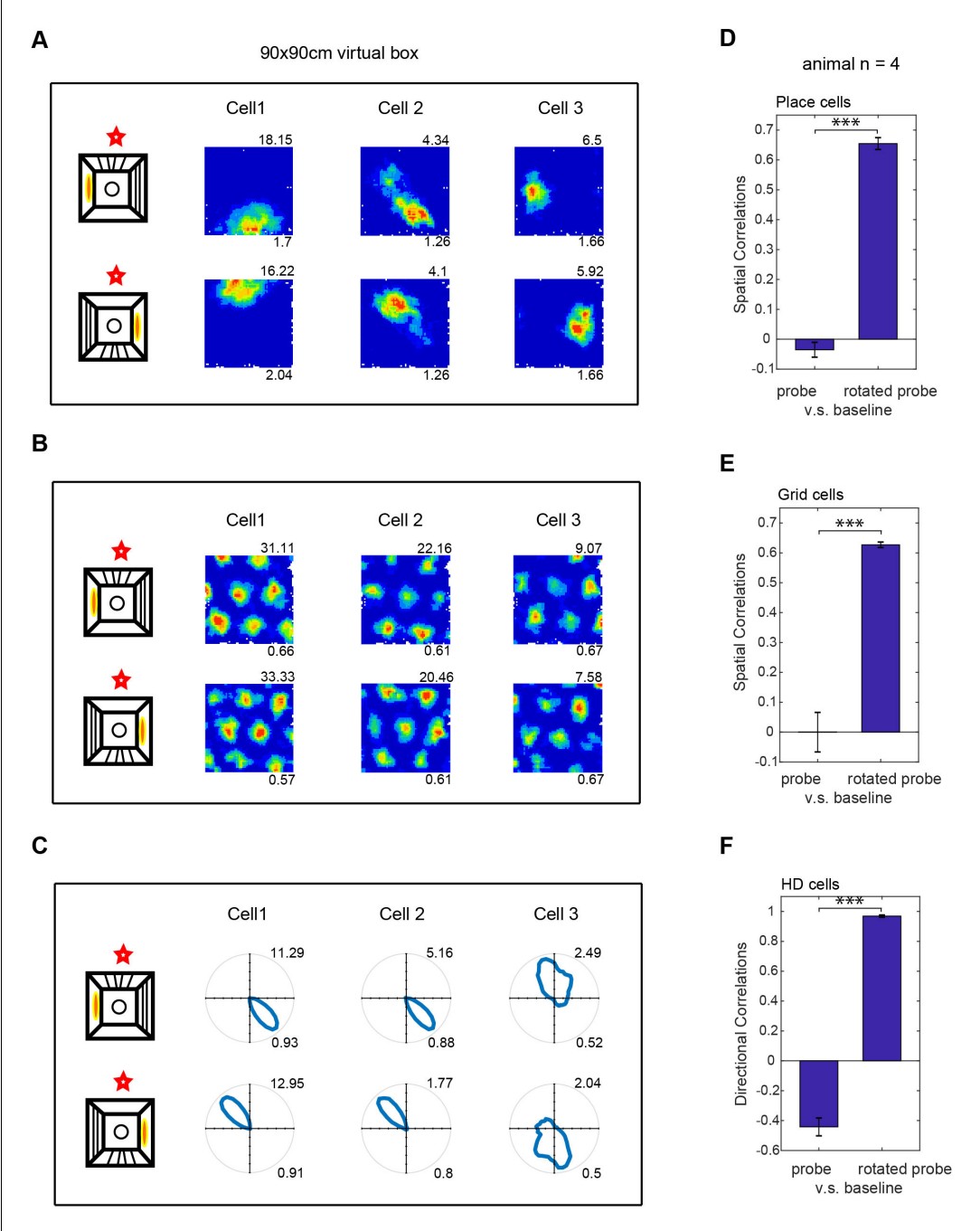

**Figure 6.** Effect of rotating the virtual environment on spatial firing patterns. (A–C) Three simultaneously recorded CA1 place cells (A), dmEC grid cells (B) and dmEC head-direction cells (C). Upper rows show firing patterns in baseline trials, lower rows show the rotated probe trials. Schematic (far let) shows the manipulation: virtual cues and entry point were rotated 180° relative to the real environment (marked by a red star). Maximum firing rates are shown top right, spatial information (A), gridness (B) or Rayleigh vector length (C) bottom right. (D–F) Spatial correlations between probe and baseline trials were significantly higher when the probe trial rate map was rotated 180° than when it was not (spatial correlations for place cells, n = 123, t(122) =19.44, p<0.001; grid cells, n = 18, t(17)=9.41, p<0.001; HD cells, n = 17, t(16)=24.77, p<0.001).
DOI: https://doi.org/10.7554/eLife.34789.016

The following figure supplement is available for figure 6:

**Figure supplement 1.** Spatial information of place cells that did not follow the 180 degree rotation of VR environment (n = 7, spatial information were 0.32 ± 0.23, 0.14 ± 0.07, 0.15 ± 0.07 and 0.16 ± 0.09 in R, VR control, VR rotated and VR control trials respectively).
DOI: https://doi.org/10.7554/eLife.34789.017

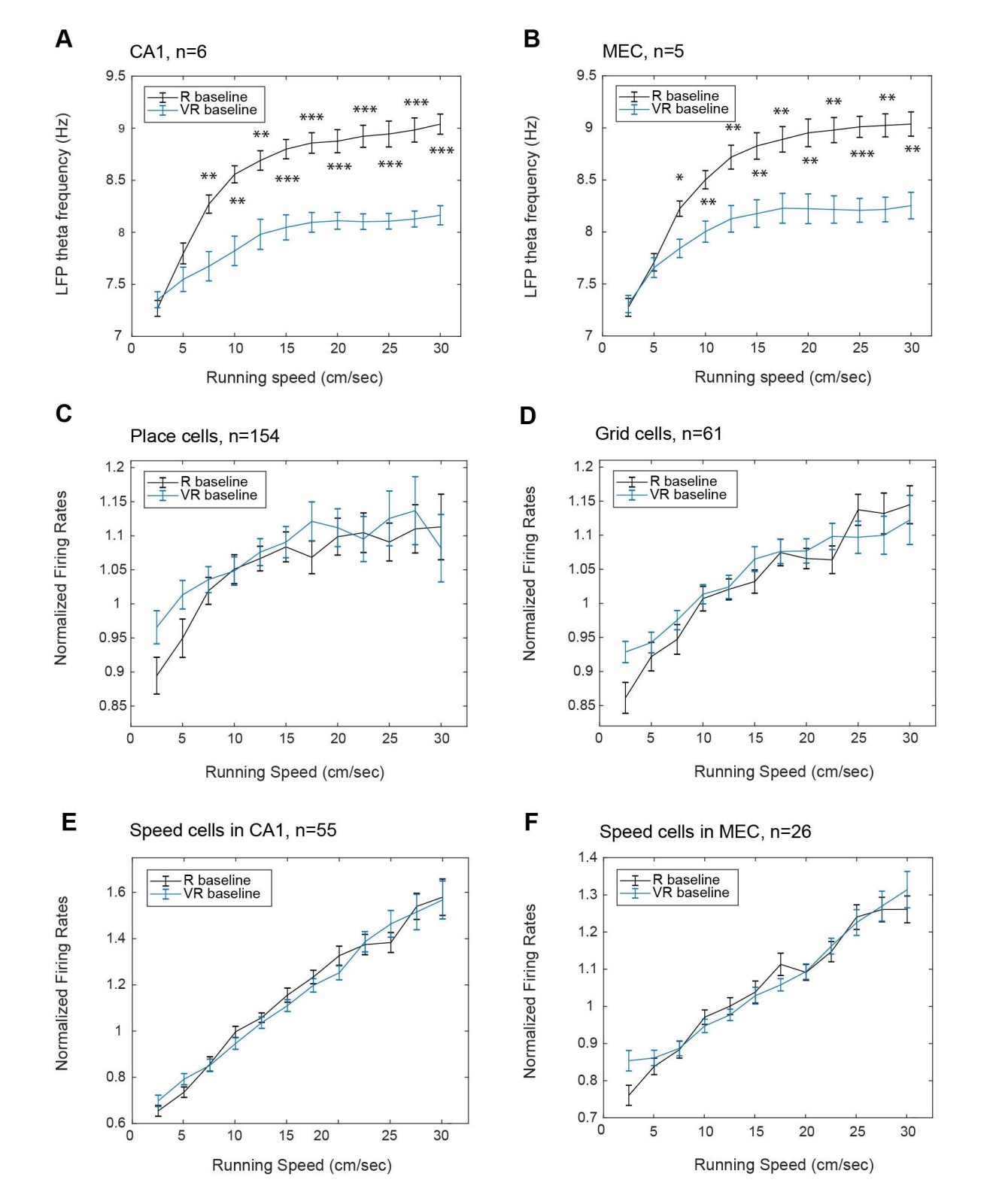

**Figure 7.** Effect of running speed on theta frequency and firing rates in real and virtual environments. Relationship between running speed in VR (blue) and R (black) on instantaneous LFP theta frequency in CA1 (A, n = 6); instantaneous LFP theta frequency in dmEC (B, n = 5); firing rates of place cells in CA1 (C, n = 154); firing rates of grid cells in dmEC (D, n = 61); speed-modulated cells in CA1 (E, n = 55); firing rates of speed-modulated cells in dmEC (F, n = 26). Lines show the mean (±s.e.m) theta frequency in each running speed bin (2.5 cm/s to 30 cm/s).

*Figure 7 continued on next page*

*Figure 7 continued*

DOI: https://doi.org/10.7554/eLife.34789.018

The following figure supplement is available for figure 7:

**Figure supplement 1.** Spatial cell field size was modulated by running speed in a similar way in R and VR environments.

DOI: https://doi.org/10.7554/eLife.34789.019

and rotating the walls and floor between trials), but reliable within-trial local cues (e.g. olfactory, tactile) may contribute to localization of firing nonetheless (*Ravassard et al., 2013*). Thus uncontrolled local cues in real experiments may be useful for supporting an orientation-independent locational response that can be bound to the distinct visual scenes observed in different directions, see also *Acharya et al., (2016)*. In this case, by removing these local cues, the use of VR leaves the locational responses of place cells more prone to modulation by the remaining (directionally specific) visual

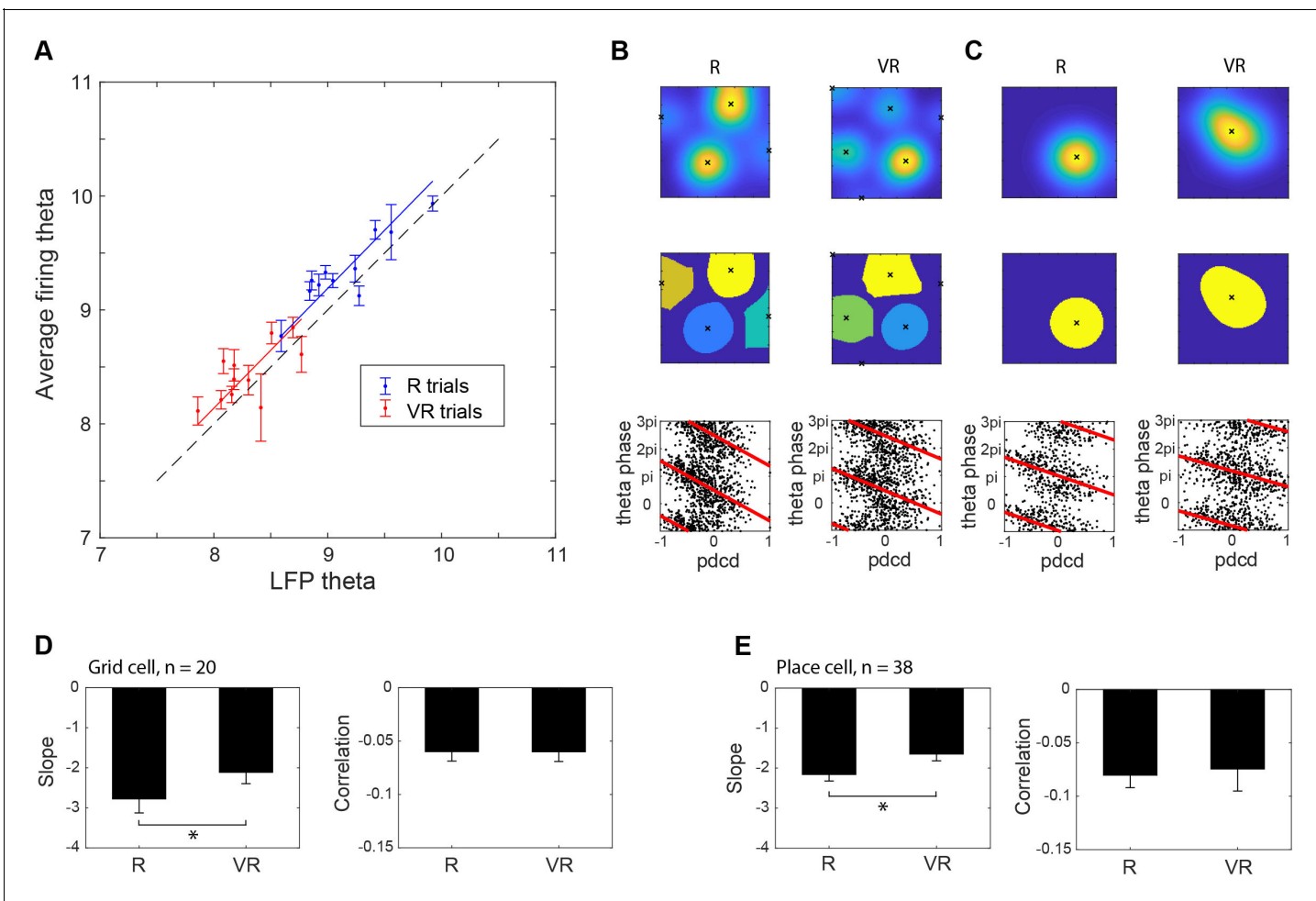

**Figure 8.** Theta phase precession. (A) Theta frequency modulation of firing rate versus LFP theta frequency (175 theta modulated cells recorded from EC and CA1, including 20 grid cells and 38 place cells). (B) Example of a grid cell's theta phase precession in real (left) and virtual environments (right), showing the detection of peaks in the smoothed firing rate map (above), division of data into separate firing fields (middle) and firing phase with respect to LFP theta plotted against distance through field along the current direction of motion (pdcd). (C) Example of a place cell's theta phase precession in real (left) and virtual environments (right), shown as in B). (D) Comparison of grid cell theta phase precession in R and in VR. Precession slope is lower in absolute value in VR than in R (n = 20, t(19)=-2.55, p<0.05). Phase-pdcd correlation strengths were comparable in VR and in R (n = 20, t(19)=0.02, p=0.98). (E) Comparison of place cell theta phase precession in R and VR. Precession slope is lower in absolute value in VR than in R (n = 38, t(37)=-2.19, p<0.05). Phase-pdcd correlation strengths were not different in VR and in R (n = 38, t(37)=-0.24, p=0.82).

DOI: https://doi.org/10.7554/eLife.34789.020

cues. We note that grid cells did not show an increase in directional modulation. This may indicate that place cell firing is more influenced by environmental sensory inputs (directional visual cues in VR), whereas grid cell firing might be more influenced by self-motion, and thus less influenced by directional visual cues. However, this would need to be verified in future work.

In conclusion, by using VR, the system presented here offers advantages over traditional paradigms by enabling manipulations that are impossible in the real-world, allowing visual projection of an environment that need not directly reflect a physical reality or the animals' movements. Differences between the firing patterns in VR and R suggest broader spatial tuning as a possible response to under-estimated translation or spatial uncertainty caused by missing or conflicting inputs, a role for local cues in supporting directionally independent place cell firing and potentially for self-motion cues in supporting directionally independent grid cell firing. Finally, the effect of moving from R to VR in reducing the dependence on running speed of the LFP theta frequency but not of neuronal firing rates suggests two distinct mechanisms for speed coding, potentially reflecting a differential dependence on linear vestibular acceleration cues. However, the temporal coding seen in theta phase precession of place (*O'Keefe and Recce, 1993*) and grid (*Hafting et al., 2008*) cell firing appeared to be fully present in VR despite the reduced theta frequency and absent linear acceleration cues (*Figure 8*).

Previous body-rotation VR systems for rats (*Aronov and Tank, 2014*; *Hölscher et al., 2005*) also allow expression of the two dimensional firing patterns of place, grid and head-direction cells. However, by working for mice and by constraining the head to rotation in the horizontal plane, our system has the potential for future use with multi-photon imaging using genetically encoded calcium indicators. The use of multiple screens and floor projectors is not as elegant as the single projector systems (*Aronov and Tank, 2014*; *Hölscher et al., 2005*) but allows the possible future inclusion of a two photon microscope above the head without interrupting the visual projection, while the effects of in-plane rotation on acquired images should be correctable in software (*Voigts and Harnett, 2018*).

## Materials and methods

### Virtual reality

A circular head-plate made of plastic (Stratasys Endur photopolymer) is chronically attached to the skull, with a central opening allowing the implant of tetrodes for electrophysiological recording (see Surgery). The head-plate makes a self-centring joint with a holder mounted in a bearing (Kaydon reali-slim bearing KA020XP0) and is clipped into place by a slider. The bearing is held over the centre of an air-supported Styrofoam ball. Four LCD screens placed vertically around the ball and two projectors onto a horizontal floor provide the projection of a virtual environment. The ball is prevented from yaw rotation to give the mouse traction to turn and to prevent any rotation of the ball about its vertical axis, following *Aronov and Tank (2014)*, (see *Figure 1A–E*).

The virtual environment runs on a Dell Precision T7500 workstation PC running Windows 7 64-bit on a Xeon X5647 2.93 GHz CPU, displayed using a combination of four Acer B236HL LCD monitors mounted vertically in a square array plus two LCD projectors (native resolution 480 × 320, 150 lumens) mounted above to project floor texture. The head-holder is at the centre of the square and 60 mm from the bottom edge of the screens, and 9500 mm below the projectors. The LCD panels are 514 mm x 293 mm, plus bezels of 15 mm all around. These six video feeds are fed by an Asus AMD Radeon 6900 graphics card and combined into a single virtual display of size 5760 × 2160 px using AMD Radeon Eyefinity software. The VR is programmed using Unity3d v5.0.2f1 which allows virtual cameras to draw on specific regions of the virtual display, with projection matrices adjusted (see Kooima, 2008 http://csc.lsu.edu/~kooima/articles/genperspective/index.html) to the physical dimensions and distances of the screens and to offset the vanishing point from the centre. For example, a virtual camera facing the X-positive direction renders its output to a portion of the virtual display which is known to correspond to the screen area of the physical monitor facing the X-negative direction.

Translation in the virtual space is controlled by two optical mice (Logitech G700s gaming mouse) mounted with orthogonal orientations at the front and side of a 200 mm diameter hollow polystyrene sphere, which floats under positive air pressure in a hemispherical well. The optical mice drive X

and Y inputs respectively by dint of their offset orientations, and gain can be controlled within the Unity software. Gain is adjusted such that real-world rotations of the sphere are calibrated so that a desired environmental size (e.g. 600 mm across) corresponds to the appropriate movement of the surface of the sphere under the mouse (i.e. moving 600 mm, or just under one rotation, on the sphere takes the mouse across the environment). Mouse pointer acceleration is disabled at operating system level to ensure movement of the sphere is detected in a linear fashion independent of running speed.

The mouse is able to freely rotate in the horizontal plane, which has no effect on the VR display (but brings different screens into view). Rotation is detected and recorded for later analysis using an Axona dacqUSB tracker which records the position of two LEDs mounted at ~25 mm offset to left and right of the head stage amplifier (see Surgery). Rotation is sampled at 50 Hz by detection of the LED locations using an overhead video camera, while virtual location is sampled and logged at 50 Hz.

Behavior is motivated by the delivery of milk rewards (SMA, Wysoy) controlled by a Labjack U3HD USB Data Acquisition device. A digital-to-analogue channel applies 5V DC to a control circuit driving a 12V Cole-Parmer 1/16' solenoid pinch valve, which is opened for 100 ms for each reward, allowing for the formation of a single drop of milk (5 uL) under gravity feed at the end of a 1/32' bore tube held within licking distance of the animal's mouth.

Control of the Labjack and of reward locations in the VR is via UDP network packets between the VR PC and a second experimenter PC, to which the Labjack is connected by USB. Software written in Python 2.7 using the Labjack, tk (graphics) and twistd (networking) libraries provide a plan-view graphical interface in which the location of the animal and reward cues in the VE can be easily monitored and reward locations manipulated with mouse clicks (see *Figure 1*).

## Animals

Subjects (14 male C57Bl/6 mice) were aged 11–14 weeks and weighed 25–30 grams at the time of surgery. Mice were housed under 12:12 inverted light-dark cycle, with lights on at 10am. All work was carried out under the Animals (Scientific Procedures) Act 1986 and according to Home Office and institutional guidelines.

## Surgery

Throughout surgery, mice were anesthetized with 2–3% isoflurane in $O_2$. Analgesia was provided pre-operatively with 0.1 mg/20 g Carprofen, and post-operatively with 0.1 mg/20 g Metacam. Custom-made head plates were affixed to the skulls using dental cement (Kemdent Simplex Rapid). Mice were implanted with custom-made microdrives (Axona, UK), loaded with 17 μm platinum-iridium tetrodes, and providing buffer amplification. Two mice were implanted with eight tetrodes in CA1 (ML: 1.8 mm, AP: 2.1 mm posterior to bregma), three mice with eight tetrodes in the dorsomedial entorhinal cortex (dmEC, ML = 3.1 mm. AP = 0.2 mm anterior to the transverse sinus, angled 4° posteriorly), and nine mice received a dual implant with one microdrive in right CA1 and one in left dmEC (each mircrodrive carried four tetrodes). After surgery, mice were placed in a heated chamber until fully recovered from the anesthetic (normally about 1 hr), and then returned to their home cages. Mice were given at least 1 week of post-operative recovery before cell screening and behavioral training started.

After experiments ended the 11 mice that had participated in the main experiment were killed with an overdose of sodium pentobarbital and perfused transcardially with saline followed by formalin solution. Brains were stored in formalin overnight before transferred to 30% sucrose solution for 2 days. Slicing were then done coronally for CA1 implanted hemisphere and sagittally for dmEC implanted hemisphere into 30-um-thick sections, which were mounted and stained using Thionin solution (*Figure 1—figure supplement 3*).

## Behavioral training

Behavioral training in the virtual reality setup started while tetrodes were approaching target brain areas (see Screening for spatial cells). Behavioral training involved four phases, VR trials lasted 40 min. Firstly, mice experienced an infinitely long 10 cm-wide virtual linear track, with 5 μL milk drops delivered as rewards. Reward locations were indicated by virtual beacons (high striped cylinders with

a black circular base, see *Figure 1—figure supplement 1A*), which were evenly placed along the track (see *Figure 1—figure supplement 1C*). When the mouse contacted the area of the base, milk was released and the beacon disappeared (reappearing in another location). The lateral movement of the mice was not registered in this phase. The aim of this training phase was to habituate the mice to being head restrained and train them to run smoothly on the air-cushioned ball. It took 3 days, on average, for mice to achieve this criterion and move to the next training phase.

During the second training phase mice experienced a similar virtual linear track (see *Figure 1— figure supplement 1B*), which was wider than the first one (30 cm wide). During this phase, reward beacons were evenly spaced along the long axis of the track, as before, but placed pseudo-randomly in one of three pre-defined positions on the lateral axis (middle, left or right). The aim of this training phase was to strengthen the association between rewards and virtual beacons, and to train animals to navigate towards rewarded locations via appropriate rotations on top of the ball. This training phase also took three days, on average.

During the third training phase, mice were introduced into a virtual square arena placed in the middle of a larger virtual room (see *Figure 1* and *Figure 1—figure supplement 1*). The virtual arena had size 60 × 60 cm or 90 x 90 cm for different mice. Reward beacons had a base of diameter that equalled to 10% of the arena width. Mice were trained on a 'random foraging' task, during which visible beacons were placed in the square box at random locations (at any given time only one beacon was visible).

The last training phase was the 'fading beacon' task. During this task, every fourth beacon occurred in a fixed location (the three intervening beacons being randomly placed within the square enclosure; see *Figure 1—figure supplement 1D*). At the beginning of this training phase the 'fixed location beacon' slowly faded from view over 10 contacts with decreasing opacity. The beacon would remain invisible as long as mice could find it, but would became visible again if mice could not locate it after 2 min of active searching. Once mice showed consistent navigation toward the fading fixed beacon, they were moved to the 'faded beacon' phase of the task where the 'fixed location beacon' was invisible from the start of the trial and remained invisible throughout the trial, with two drops of milk given as reward for contact. This trial phase therefore requires mice to navigate to an unmarked virtual location starting from different starting points (random locations where the third visible beacon was placed). As such, the 'fading beacon' task serves like a continuous version of a Morris Water Maze task (*Voigts and Harnett, 2018*), combining reference memory for an unmarked location with a foraging task designed to optimise environmental coverage for the assessment of spatial firing patterns. Mice typically experienced one trial per day.

## Behavioral analyses

During electrophysiological recording in the real environment (R), the mouse's position and head orientation were tracked by an overhead camera (50 Hz sampling rate) using two infra-red LEDs attached to the micro-drive at a fixed angle and spacing (5 cm apart). Brief losses of LED data due to cable obstruction (typically affecting a single LED for <500 ms) were corrected with linear interpolation between known position values. Interpolation was carried out for each LED separately. The position values for each LED were then smoothed, separately, using a 400 ms boxcar filter. During electrophysiological recording in VR, head orientation was tracked as in R, the path, running speed and running direction was inferred from the VR log at 50 Hz (movements of VR location being driven by the computer mice tracking the rotation of the ball, see above).

Path excess ratio was defined as the ratio between the length of the actual path that an animal takes to run from one reward location to another, and the distance between the two reward locations. All training trials in the VR square and the real (R) square environments from the 11 mice in the main experiment were included in the behavioral analyses in *Figures 1–2*.

## Screening for spatial cells

Following recovery, mice were food restricted to 85% of their free-feeding body weight. They were then exposed to a recording arena every day (20 min/day) and screening for neural activity took place. The recording arena was a 60 × 60 cm square box placed on a black Trespa 'Toplab' surface (Trespa International B.V., Weert, Netherlands), and surrounded by a circular set of black curtains. A white cue-card (A0, 84 × 119 cm), illuminated by a 40 W lamp, was the only directionally polarizing

cue within the black curtains. Milk (SMA Wysoy) was delivered as drops on the floor from a syringe as rewards to encourage foraging behavior. Tetrodes were lowered by 62.5 µm each day, until grid or place cell activity was identified, in dmEC or CA1 respectively. Neural activity was recorded using DACQ (Axona Ltd., UK) while animals were foraging in the square environment. For further details see *Chen et al. (2013)*.

## Recording spatial cell activity

Each recording session consisted of at least one 40 min random-foraging trial in a virtual reality (VR) square environment (see above for behavioral training). For seven mice, the virtual environment had size 60 × 60 cm and for 4 mice 90 × 90 cm when recording took place. After one (or more) 40-min random-foraging trials in the virtual square, mice were placed in a real-world square ('R', 60 × 60 cm square, similar to the screening environment, see *Figure 3—figure supplement 2A*) for a 20-min random-foraging trial in real world.

Additionally, four mice also underwent a virtual cue rotation experiment, which consisted of two 40-min random-foraging VR trial (one baseline VR trial and one rotated VR trial) and one 20-min R trial. Two mice navigating 60 × 60 cm VR squares and two 90 × 90 cm squares participated in this experiment. In the rotated VR trials, all cues in the virtual reality environment rotated 180 degrees compared to the baseline trial, as was the entry point mice were carried into the VR rig from.

In order to control for sequential effect between VR and R trials, we performed additional recordings from four mice (three new to the experiment, one that had performed the main experiment). On one day they experienced a trial in the R square before a trial in the VR square (60 × 60 cm) environment. On the next day the VR trial preceded the R trial. We also introduced those four mice to a novel cylindrical VR and cylindrical R environment which shared similar wall and floor visual patterns. Four trials were recorded on the same day in the order: VR cylinder, VR square, R cylinder and R square. All VR trials were 40 min and R trials 20 min in length (see *Figure 3—figure supplement 2* and *Figure 4—figure supplement 2*).

## Firing rate map construction and spatial cell classification

Spike sorting was performed offline using an automated clustering algorithm (KlustaKwik [*Kadir et al., 2014*]) followed by a manual review and editing step using an interactive graphical tool (waveform, Daniel Manson, http://d1manson.github.io/waveform/). After spike sorting, firing rate maps were constructed by binning animals' positions into 1.5 × 1.5 cm bins, assigning spikes to each bin, smoothing both position maps and spike maps separately using a 5 × 5 boxcar filter, and finally dividing the smoothed spike maps by the smoothed position maps.

Cells were classified as place cells if their spatial information in baseline trials exceeded the 99th percentile of a 1000 shuffled distribution of spatial information scores calculated from rate maps where spike times were randomly offset relative to position by at least 4 s. Cells were classified as grid cells if their gridness scores in baseline trials exceeded the 99th percentile of a shuffled distribution of 1000 gridness scores (*Burgess et al., 2005*). Place and grid cells additionally required a peak firing rate above 2 Hz for clarity of classification. Cells were classified as head direction cells if their Rayleigh vector lengths in baseline trials exceeded the threshold of the 99th percentile population shuffling (this sample included two conjunctive grid x direction cells (*Sargolini et al., 2006*, see *Figure 5—figure supplement 1B*).

Speed-modulated cells were classified from the general population of the recorded cells following *McNaughton et al. (1983)*. Briefly, the degree of speed modulation for each cell was characterized by first defining the instantaneous firing rate of the cell as the number of spikes occurring in each position bin divided by the sampling duration (0.02 s). Then, a linear correlation was computed between the running speeds and firing rates across all position samples in a trial, and the resulting r-value was taken to characterize the degree of speed modulation for the cell. To be defined as speed-modulated, the r-value for a cell had to exceed the 99th percentile of a distribution of 1000 r-values obtained from spike shuffled data.

When assessing the directional modulation of place and grid cell firing (*Figures 3* and *4*), apparent directional modulation can arise in binned firing rate data from heterogenous sampling of directions within the spatial firing field (*Burgess et al., 2005*; *O'Keefe and Recce, 1993*). Accordingly we fit a joint ('pxd') model of combined place and directional modulation to the data (maximising the

likelihood of the data [*Burgess et al., 2005*]) and perform analyses on the directional model in addition to the binned firing rate data.

Theta modulation of cell firing during the main experiment were computed using Maximum likelihood estimation of the distribution of lags, following *Climer et al. (2015)*. Cells with theta index higher than the confidence interval of 95% were classified as theta rhythmic cells. Among that population, two-dimensional phase precession was estimated for cells that were also classified as place cells or grid cells, following *Climer et al. (2015)*. In brief, each running trajectory passing through the defined place field was normalized and mapped radially on an unit circle so that the proportional distance of the animal between the field edge and peak was preserved and so that the average running direction was zero (from left to right). The distance of the animal from the peak projected onto the instantaneous running direction ('Pdcd') was calculated, representing the distance the animal has travelled through the field (range −1 to 1). The theta phase of each spike was computed using the Hilbert transform of the smoothed LFP.

## Acknowledgements

We acknowledge support from the Wellcome Trust, European Union's Horizon 2020 research and innovation programme (grant agreement No. 720270, Human Brain Project SGA1), Biotechnology and Biological Sciences Research Council, European Research Council and China Scholarship Council, and technical help from Peter Bryan, Duncan Farquharson, Daniel Bush and Daniel Manson.

## Additional information

### Competing interests

Neil Burgess: Reviewing editor, *eLife*. The other authors declare that no competing interests exist.

### Funding

| Funder | Grant reference number | Author |
| --- | --- | --- |
| Wellcome | 202805/Z/16/Z | Neil Burgess |
| Horizon 2020 Framework Programme | Research and Innovation program 720270 | Guifen Chen Francesca Cacucci Neil Burgess |
| Biotechnology and Biological Sciences Research Council | BB/I021221/1 | Francesca Cacucci |
| H2020 European Research Council | DEVSPACE Starting grant | Francesca Cacucci |
| China Scholarship Council | 201509110138 | Yi Lu |

The funders had no role in study design, data collection and interpretation, or the decision to submit the work for publication.

### Author contributions

Guifen Chen, Conceptualization, Formal analysis, Supervision, Investigation, Visualization, Methodology, Writing—review and editing; John Andrew King, Francesca Cacucci, Conceptualization, Resources, Software, Supervision, Methodology, Project administration, Writing—review and editing; Yi Lu, Formal analysis, Investigation, Visualization, Methodology, Writing—review and editing; Neil Burgess, Conceptualization, Resources, Formal analysis, Supervision, Funding acquisition, Investigation, Visualization, Methodology, Writing—original draft, Project administration, Writing—review and editing, Physical design and construction

### Author ORCIDs

John Andrew King http://orcid.org/0000-0002-0269-3790
Yi Lu http://orcid.org/0000-0002-4320-675X
Neil Burgess http://orcid.org/0000-0003-0646-6584

## Ethics

Animal experimentation: All work was carried out under the Animals (Scientific Procedures) Act 1986 and according to Home Office and institutional guidelines.

## Decision letter and Author response

Decision letter https://doi.org/10.7554/eLife.34789.025
Author response https://doi.org/10.7554/eLife.34789.026

## Additional files

### Supplementary files

• Transparent reporting form
DOI: https://doi.org/10.7554/eLife.34789.021

### Data availability

Data have been made available via the Open Science Framework platform (https://osf.io/yvmf4/)

The following dataset was generated:

| Author(s) | Year | Dataset title | Dataset URL | Database, license, and accessibility information |
|---|---|---|---|---|
| Francesca Cacucci, Neil Burgess | 2018 | Data from Spatial cell firing during virtual navigation of open arenas by head-restrained mice | https://osf.io/yvmf4/ | Publicly available on the Open Science Framework |

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
