## [Decision Letter]

Thank you for sending your article entitled "Spatial cell firing during virtual navigation of open arenas by head-restrained mice" for peer review at *eLife*. Your article is being evaluated by three peer reviewers, and the evaluation is being overseen by a Reviewing Editor and Michael Frank as the Senior Editor.

Given the list of essential revisions, including new experiments, the editors and reviewers invite you to respond within the next two weeks with an action plan and timetable for the completion of the additional work. We plan to share your responses with the reviewers and then issue a binding recommendation.

All reviewers found many positive aspects about the paper. However, the reviewers raised major concerns about your claims regarding neural activity differences between virtual reality and real world environments, an important and crucial component of the work. Reviewers felt that the reported differences are not strongly supported by the results (in their present form) because of potential confounds such as fixed sequence of the two experiences (i.e., virtual reality was always experienced before real world), richness of cues in the testing environments (i.e., virtual reality was a cue-rich environment; real world was surrounded by curtains and only included a single cue), etc. A complete list of reviewers' concerns is provided below.

*Reviewer #1:*

In the current manuscript, the authors demonstrate a mouse virtual reality (VR) system, which allows head-restrained mice to navigate in a two-dimensional (2D) environment. The key new feature is a bearing attached to an implanted head-plate that allows the mouse's head to be restrained, but able to rotate around a vertical axis on the surface of a spherical treadmill; self-induced rotations in space are possible using this system, while maintaining a level of head restraint that would facilitate the use of optical measurements of neural activity in the future. The authors successfully recorded characteristic firing patterns of place cells in the hippocampus and grid and head direction cells in the medial entorhinal cortex (MEC), and they compared the activity features of these cells with those in real (R) environments. Overall, the development of the mouse 2D VR is an advance for the field, as it allows precise control of animal's environment and is potentially compatible with two-photon based calcium imaging of neural population activity. However, some conclusions of the study, in particular the comparisons between R and VR, are less supported and require more analyses. In addition, the writing of the manuscript needs to be improved, as explained below.

1) Analysis of VR electrophysiology during behavior

The behaviors studied in the R and VR environments are distinct. In R, animals are randomly foraging based on visual and olfactory cues, whereas in VR animals are either randomly foraging based on visual cues or simply navigating to a fixed location. Random foraging and navigating to a fixed location are very different behaviors. It is surprising that the electrophysiology from these two VR behavioral modes was not analyzed separately, given that distinct computations might support distinct behaviors. It is possible that if analyzed separately, the two similar behaviors (random foraging in R vs. VR) would have more similar electrophysiological profiles, and that the non-random navigation in VR accounts for the entirety of the difference in R vs. VR. In addition, the authors should make clear in the Results what behavioral paradigm is being used. Is it random foraging + morris water maze in VR, or just random foraging, etc.?

Finally, authors should motivate the choice to use two differently-sized boxes for recording and clarify when data were pooled from both boxes or separately analyzed?

2) VR vs. R interpretation

The authors suggest different mechanisms for the observed differences in activity between R and VR in the Abstract. However, there are at least three types of concerns about R vs. VR comparisons, detailed below:

Visual differences between R and VR

The virtual reality and real environments are visually completely different (VR: Cue-rich square with multiple wall textures, colors, and encompassing environment vs. R: square box with black curtains and white cue card as only polarizing feature, subsection “Screening for spatial cells”). How do the authors know that the differing behavioral and electrophysiological results are due to VR vs. R, or translational vestibular inputs, rather than the large difference in visual environments? The authors should specify in the Abstract that the observed phenomenological differences apply to this particular R vs. VR setup. Additionally, the authors should discuss the possibility of confounds in their interpretation (e.g. due to visual differences in their R vs. VR environments) in the Discussion.

Residual impact of real world on VR recordings

For example, if expansion of field sizes reflected spatial uncertainty caused by the conflict between uncontrolled distal cues with virtual cues, was there any clue in the current data suggesting this conflict? It would be interesting to know during virtual navigation what percentages of cells in hippocampus and dmEC were still responding to cues in the real environment (classified as place or grid cells using the coordinates of the real environment). For the place/grid/head direction cells, which were classified in the real environment but not the virtual environment, what did their activity patterns in the virtual environment look like? When the virtual environment was rotated 180°, what percentage of cells followed and did not follow this rotation? Conclusions of these analyses may reveal the potential conflict.

Missing local cues in VR

The authors state that 'omni-directional place cell firing in R reflects local cues unavailable in VR'. This conclusion would be much more believable if this directionality were observed in at least a different, additional VR environment. If the authors recorded cells in a different VR, did they see the same increase of directionality of place cells? Otherwise, is it possible that the directionality in VR was due to this particular VR design? Furthermore, could the authors clarify what they mean by 'local cues' – do they refer to olfactory or tactile cues?

In light of these potential caveats, it appears inappropriate to state the suggested mechanisms comprising the second half of the Abstract without further analysis. The authors should either provide more evidence, or temper/remove these claims from the Abstract.

3) Tetrode placement

Histological data verifying tetrode placement is the standard for this type of experiment, but it is not provided in the manuscript.

4) Fraction of place cells

The fraction of place cells in CA1 in this manuscript is ~75%, which seems unusually high. Could the authors provide commentary about the reasons for this?

*Reviewer #2:*

This manuscript demonstrates that place/grid cells can be recorded from head-fixed mice in a virtual reality preparation, provided that the animal can rotate its head on a linear plane. They compared the place/firing fields of these place/grid cells with those recorded in equivalent real environments. They observed two major differences. Firstly, the place/firing fields of both place and grid cells expanded to a similar degree, approximately 1.4 times relative to the real environment. Although the speed of the animals modulated the firing rate these cells, the frequency of theta oscillations exhibited a weaker speed modulation in the virtual environment. Therefore, the vestibular inputs related to linear acceleration affected the theta oscillatory system whereas the spatially firing cells used other means for speed coding. As the authors point out, it is possible that spatial expansion is related to the weaker speed modulation of theta oscillations. The work is important for two reasons: firstly it introduces a new technique to study the activity of place coding cells in virtual two-dimensional environments with a potential to use it in head-fixed imaging preparations as well. Secondly, it demonstrates that speed encoding mechanism of spatial firing cells and those of theta oscillations are different and it implies that the lack of linear acceleratory vestibular inputs leads to the expansion of firing fields of both place and grid cells. Both of these are important for the understanding the mechanism behind the firing of place/grid cells. I have only a few questions and suggestions.

1) Not sure whether place/firing field sizes of place/grid cells is modulated by speed in either the real or virtual environments. If so, comparing firing fields at speed ranges associated with similar theta frequency in both preparations could give more insight as to whether place field expansion and theta frequency speed modulation are related.

2) The firing of place and grid cells are influenced to a large degree by theta oscillation. In particular, they fire periodically with a slightly faster frequency than theta oscillations, a property enabling theta phase precession. Given that in speed modulation these cells acted differently from theta oscillation, one may wonder whether their periodic firing may also be different in a virtual environment to that of the field oscillations. So it is possible that their periodic modulation may not follow the theta frequency fluctuations with a consistently faster periodicity. I understand that cells may not fire a sufficient number of action potentials for such an analysis but perhaps the analysis could be performed for some broader speed ranges or simply checking whether the periodicity difference of the autocorrelations and those of detected theta waves on average show similar consistent differences for the real and virtual environments. If so, this would imply impairment in two-dimensional phase precession and suggest that place cells can be modulated with both vestibular dependent and independent inputs.

*Reviewer #3:*

The authors demonstrate a new head-fixed method that can be used with mice to study behavior in VR with the freedom of horizontal head and body movement. This is a novel set up, and in many ways is an improvement on traditional head-fixed methods in VR. For example, the authors show they can observe head-direction cells and 2D place fields that are similar to real worlds. As the authors discuss, VR brings many benefits to experimenters, by providing greater control over the animal's experience. The author's novel head-fixed method brings the VR experience closer to a real-world experience, without any loss in experimenter control. For these reasons, I believe this is a useful method for the study of spatial representations, and other processes, during well-controlled behavior. But I do have some concerns that I believe should be addressed before publication of this paper (see below).

In their Materials and methods section it is reported that on each recording day mice always experienced VR followed by R (subsection “Recording spatial cell activity”). This is a potential confound. It would be a good idea to reverse the order to determine whether any differences in spatial representations that are reported in this paper are caused by the sequence of the two experiences. This is important, as a major component of this paper is that certain measures of place and grid fields are different in VR vs R. Fatigue and attention could affect spatial representations, and these processes change over time. Therefore the time relationship should be taken into consideration before any major conclusions are made. On this note, the authors also state that the VR session was always twice as long as the R session (40 min vs. 20 min, respectively). To avoid any potential confounds that differences in session length might bring to the data, it would be best to compare data from the first 20 min in VR to the 20 min session in R.

Could this setup really be used with multi photon imaging? It's hard to imagine this would work given the major rotational movement of the brain. In theory, it is possible, but in practice it would be very difficult. It is potentially misleading to make this claim without evidence. The authors do say "potentially compatible with 2P imaging", which is good, but I think they should discuss how to implement it in more detail than they have, maybe by discussing the problems that would need to be overcome, in lieu of showing evidence that it works.

The% of place cells reported by the authors in CA1 is very high: 186/231 in VR is 81%, and 175/231 in R is 76%. Most studies have found approximately 30% of CA1 cells have significant place fields. Why is the% reported here so different? This should be discussed.

What about phase precession in R vs. VR? The authors show that running speed vs. theta frequency is different in R vs. VR, but spike rates in PFs vs. running speed are similar in R vs. VR, which would suggest differences in phase precession between the two experiences. The authors should have this data, so it would be a nice addition to the paper if they analyzed it and presented here. I think readers of this paper would want to know about this.

[Editors' note: the authors’ plan for revisions was approved and the authors made a formal revised submission.]

---

## [Author Response]

Reviewer #1:[…] However, some conclusions of the study, in particular the comparisons between R and VR, are less supported and require more analyses. In addition, the writing of the manuscript needs to be improved, as explained below.1) Analysis of VR electrophysiology during behaviorThe behaviors studied in the R and VR environments are distinct. In R, animals are randomly foraging based on visual and olfactory cues, whereas in VR animals are either randomly foraging based on visual cues or simply navigating to a fixed location. Random foraging and navigating to a fixed location are very different behaviors. It is surprising that the electrophysiology from these two VR behavioral modes was not analyzed separately, given that distinct computations might support distinct behaviors. It is possible that if analyzed separately, the two similar behaviors (random foraging in R vs. VR) would have more similar electrophysiological profiles, and that the non-random navigation in VR accounts for the entirety of the difference in R vs. VR. In addition, the authors should make clear in the Results what behavioral paradigm is being used. Is it random foraging + morris water maze in VR, or just random foraging, etc.?

All recordings presented were from the random foraging task (in R and in VR) – this was mentioned in the Materials and methods (subsections “Screening for spatial cells” and “Recording spatial cell activity”, first paragraph) and is now clarified in the second paragraph of the subsection “Electrophysiology”.

Finally, authors should motivate the choice to use two differently-sized boxes for recording and clarify when data were pooled from both boxes or separately analyzed?

We explored different sizes for the virtual environment during development of the paradigm. All data were presented pooled across sizes. We have now performed separate analyses for trials in 60x60cm and 90x90cm VR environments and their respective within-session real-world (R) trials (Figure 4—figure supplement 1). For place cells, similar results were seen irrespective of whether recordings took place in the 60x60cm or 90x90cm VR environments (e.g., the place field expansion factor being 1.44 in 90cm, 1.43 in 60cm, p=0.66), referred to in the third paragraph of the subsection “Electrophysiology”. For grid cells, similar results were also seen irrespective of whether recordings took place in the 60x60cm or 90x90cm VR environments (e.g., the grid scale expansion factor being 1.43 in 60cm, 1.36 in 90cm, p=0.78), although there were minor differences (the reduction in spatial information only reaching significance in the 60x60cm VR and that in directional information only reaching significance in the 90x90cm VR), referred to in the fifth paragraph of the aforementioned subsection.

2) VR vs. R interpretationThe authors suggest different mechanisms for the observed differences in activity between R and VR in the Abstract. However, there are at least three types of concerns about R vs. VR comparisons, detailed below:Visual differences between R and VRThe virtual reality and real environments are visually completely different (VR: Cue-rich square with multiple wall textures, colors, and encompassing environment vs. R: square box with black curtains and white cue card as only polarizing feature, subsection “Screening for spatial cells”). How do the authors know that the differing behavioral and electrophysiological results are due to VR vs. R, or translational vestibular inputs, rather than the large difference in visual environments? The authors should specify in the Abstract that the observed phenomenological differences apply to this particular R vs. VR setup. Additionally, the authors should discuss the possibility of confounds in their interpretation (e.g. due to visual differences in their R vs. VR environments) in the Discussion.

The walls of the R square box was patterned and coloured similarly to the VR environment, we apologise for the lack of clarity. A picture has been added (Figure 3—figure supplement 2A). However, the point remains valid that other (visual) differences between the R and VR environments could contribute to the differences observed in neural firing. To address this we performed further recordings in a second pair of (cylindrical) virtual and real environments – to check that differences between virtual and real environments are general to both square and cylindrical environments (Figure 3—figure supplement 2). This has been acknowledged, and discussed in the fourth paragraph of the subsection “Electrophysiology” and in the seventh paragraph of the Discussion.

Residual impact of real world on VR recordingsFor example, if expansion of field sizes reflected spatial uncertainty caused by the conflict between uncontrolled distal cues with virtual cues, was there any clue in the current data suggesting this conflict? It would be interesting to know during virtual navigation what percentages of cells in hippocampus and dmEC were still responding to cues in the real environment (classified as place or grid cells using the coordinates of the real environment). For the place/grid/head direction cells, which were classified in the real environment but not the virtual environment, what did their activity patterns in the virtual environment look like? When the virtual environment was rotated 180°, what percentage of cells followed and did not follow this rotation? Conclusions of these analyses may reveal the potential conflict.

We have examined the virtual firing patterns of cells classified as place, grid or head-direction cells in R but not in VR. In general, they have weaker spatial tuning in VR than R (as would be expected given the classifications) but we saw no specific pattern to this. For example it is not that the spatial tuning is present on a time-windowed spatial auto-correlogram but drifts over time and so does not appear in the full rate map. We have also investigated the firing of cells in the 180° rotation of the virtual environment (and the entry point to it). All head-direction cells and grid cells did follow the rotation. For place cells, 7/141 (5%), failed to follow the rotation. These place cells show much lower spatial information scores in both the R and VR conditions (see Figure 6—figure supplement 1), indicating that their lack of rotation might be the result of their weaker or less stable spatial tuning to the proximal environmental cues that were rotated. We have added reference to these results in the ninth paragraph of the subsection “Electrophysiology”.

We were not able to classify firing in VR trials as place or grid cells using the coordinates of the real environment, because the mouse’s head only occupies one real-world location during these trials (and this location was within the real recording environment).

Missing local cues in VRThe authors state that 'omni-directional place cell firing in R reflects local cues unavailable in VR'. This conclusion would be much more believable if this directionality were observed in at least a different, additional VR environment. If the authors recorded cells in a different VR, did they see the same increase of directionality of place cells? Otherwise, is it possible that the directionality in VR was due to this particular VR design? Furthermore, could the authors clarify what they mean by 'local cues' – do they refer to olfactory or tactile cues?

We investigated further whether the increased directionality of place cell firing in VR was specific to the square VR environment, by performing additional recordings of both place and grid cells while animals foraged in (visually similar) cylindrical and square R and VR environments (in 4 mice, 3 new to the experiment, yielding a total of 90 place and 9 grid cells). The increased directionality of place cells but not grid cells in VR was present in both cylinder and square environments, supporting the generality of the result (see Figure 3—figure supplement 2). Referred to in the fourth paragraph of the subsection “Electrophysiology”.

We have clarified the intended meaning of local cues (Discussion, seventh paragraph) – uncontrolled cues to location that are stable within one trial (but may well not be from trial to trial, as the apparatus is often cleaned and

swapped/rotated between trials): olfactory, tactile. Notably, these local cues can be sensed independently of the orientation of the animal, unlike visual cues.

In light of these potential caveats, it appears inappropriate to state the suggested mechanisms comprising the second half of the Abstract without further analysis. The authors should either provide more evidence, or temper/remove these claims from the Abstract.

We appreciate this point and have clarified the meaning of local cues (Discussion, seventh paragraph), added the data from cylindrical trials (Figure 3—figure supplement 2), and tempered our claims (‘may require’ – Abstract, ‘may indicate’ – Discussion).

3) Tetrode placementHistological data verifying tetrode placement is the standard for this type of experiment, but it is not provided in the manuscript.

We now provide histological verification of electrode locations in all the 11 animals in the main experiment (Figure 5—figure supplement 1). As noted in our plan of action, we are not able to provide histology for the 3 new animals used.

4) Fraction of place cellsThe fraction of place cells in CA1 in this manuscript is ~75%, which seems unusually high. Could the authors provide commentary about the reasons for this?

The proportion of 75% place cells is fairly usual as a proportion of the active cells recorded in dorsal CA1 during foraging in a specific environment. See e.g. (Wills et al., 2010; Langston et al., 2010) for similar proportions in adult rats. It is also true that the proportion of place cells that are active in any specific environment is around 30% of all cells (e.g. Guzowski et al., 1999) – but the other 70% of cells are not active (and thus we cannot estimate their number on the basis of in vivo electrophysiology).

Reviewer #2:[…] 1) Not sure whether place/firing field sizes of place/grid cells is modulated by speed in either the real or virtual environments. If so, comparing firing fields at speed ranges associated with similar theta frequency in both preparations could give more insight as to whether place field expansion and theta frequency speed modulation are related.

We have investigated the size of place and grid cell firing fields as a function of running speed in R and VR (Figure 7—figure supplement 1). There is a modulation of field size by running speed, which is similar in R and VR environments.

However this modulation is too small and non-monotonic to be able to explain differences in field size across R and VR in terms of differences in theta frequency. Thus, although theta frequency at 3-13 cm/s in R is approximately the same as theta frequency at 23-33cm/s in VR, field sizes are larger in VR than in R in both low and high speed bands. This is referred to in the tenth paragraph of the subsection “Electrophysiology”.

2) The firing of place and grid cells are influenced to a large degree by theta oscillation. In particular, they fire periodically with a slightly faster frequency than theta oscillations, a property enabling theta phase precession. Given that in speed modulation these cells acted differently from theta oscillation, one may wonder whether their periodic firing may also be different in a virtual environment to that of the field oscillations. So it is possible that their periodic modulation may not follow the theta frequency fluctuations with a consistently faster periodicity. I understand that cells may not fire a sufficient number of action potentials for such an analysis but perhaps the analysis could be performed for some broader speed ranges or simply checking whether the periodicity difference of the autocorrelations and those of detected theta waves on average show similar consistent differences for the real and virtual environments. If so, this would imply impairment in two-dimensional phase precession and suggest that place cells can be modulated with both vestibular dependent and independent inputs.

We have determined the theta-band frequency modulation of firing of theta-modulated cells and compared them to the LFP theta frequency in R and VR (new Figure 8A). This shows that firing frequency is slightly higher than the LFP the frequency, in R and VR despite the lower overall frequencies in VR, consistent with theta phase precession. We also analysed phase precession in theta-modulated place and grid cells – showing normal phase precession in VR as in R (with a lower slope consistent with the larger firing fields in VR, but the correlation between slope and field size only reaching significance for place fields in R). These results indicate that theta phase precession in place and grid cells is independent of linear vestibular acceleration signals and the absolute value of theta frequency. Text has been added in the last paragraph of the Results subsection “Electrophysiology”, in the eighth paragraph of the Discussion, and in the Abstract. Methods are included in the last paragraph of the subsection “Firing rate map construction and spatial cell classification”.

Reviewer #3:[…] In their Materials and methods section it is reported that on each recording day mice always experienced VR followed by R (subsection “Recording spatial cell activity”). This is a potential confound. It would be a good idea to reverse the order to determine whether any differences in spatial representations that are reported in this paper are caused by the sequence of the two experiences. This is important, as a major component of this paper is that certain measures of place and grid fields are different in VR vs R. Fatigue and attention could affect spatial representations, and these processes change over time. Therefore the time relationship should be taken into consideration before any major conclusions are made. On this note, the authors also state that the VR session was always twice as long as the R session (40 min vs. 20 min, respectively). To avoid any potential confounds that differences in session length might bring to the data, it would be best to compare data from the first 20 min in VR to the 20 min session in R.

We ran VR trials first as we were anxious to have well-motived animals in our initial attempts to get 2d VR to work, and because previous experiments, including our own, show that effects of trial order on spatial firing patterns are small (see e.g. grid cells in mice, Brun et al., 2008; Carpenter et al., 2017; place cells in mice Nakazawa et al., 2002).

To check whether any differences between R and VR could reflect the trial order (VR before R), we have now recorded additional data from place and grid cells in R and VR on days in which R trials both preceded and followed VR trials (in 4 mice, 3 new to the experiment). We also included analysis of the first 20 mins of VR trials (matching the length of R trials, see Figure 4—figure supplement 2). Under these conditions, the firing properties of grid and place cells are broadly similar to those shown in Figures 3 and 4, indicating the generality of these findings (subsection “Electrophysiology”, seventh paragraph).

However, the 20 grid cells in this group did show lower gridness scores in VR than R, and 43 cells were classified as grid cells in R but only 24 as grid cells in VR. Thus grid cell firing patterns can be sensitive to the use of VR and the inherent conflict between virtual and uncontrolled cues to translation. The extra sensitivity in the second group of animals might reflect their greater age at test (mice with grid cells, main experiment: n=8, age=25.4 ± 4.3 weeks; additional experiment: n=3, age=40.1 ± 11.2 weeks; t(9)=-3.34, p<.01) but this would require further verification (see the aforementioned paragraph)).

Could this setup really be used with multi photon imaging? It's hard to imagine this would work given the major rotational movement of the brain. In theory, it is possible, but in practice it would be very difficult. It is potentially misleading to make this claim without evidence. The authors do say "potentially compatible with 2P imaging", which is good, but I think they should discuss how to implement it in more detail than they have, maybe by discussing the problems that would need to be overcome, in lieu of showing evidence that it works.

We agree that the issue of feasibility could be discussed further. The problem is to build rotation correction into the processing pipeline in the same way that translation (both rigid body and locally varying) is corrected (e.g. Pnevmatikakis et al., 2016; Pachitariu et al., 2017). However, by recording the mouse’s head rotation directly, it is straightforward to remove the rigid body rotation, so that the remaining motion can be attacked with existing methods. A reasonably standard setup for 2 photon calcium imaging would be acquisition of 200μm square images at 30Hz. So the problem comes down to how much rotation occurs within the acquisition of each frame. The mean rotation rate (Figure 1N) is 25^o^/s or < 1^o^/frame, equivalent to a systematic displacement of each scan line that is < 2μm (~ 100μm x π/180 rads towards the ends). As this is well within the brain movement currently compensated for (e.g. Kong et al., 2016) we see this as feasible. Indeed a recent paper on bioRxiv (Voigts and Harnett, 2018) demonstrates a solution to exactly this problem – we have now added a reference to the end of the Discussion to point this out.

The% of place cells reported by the authors in CA1 is very high: 186/231 in VR is 81%, and 175/231 in R is 76%. Most studies have found approximately 30% of CA1 cells have significant place fields. Why is the% reported here so different? This should be discussed.

Please see answer to reviewer 1’s point 4.

What about phase precession in R vs. VR? The authors show that running speed vs. theta frequency is different in R vs. VR, but spike rates in PFs vs. running speed are similar in R vs. VR, which would suggest differences in phase precession between the two experiences. The authors should have this data, so it would be a nice addition to the paper if they analyzed it and presented here. I think readers of this paper would want to know about this.

Phase precession in 2d is relatively hard to analyse even in rats, requiring lots of spikes, clear LFP theta and a rule for combining data from different types of trajectories. And the theta modulation of firing is weaker in mice than in rats in our experience. However, we have now also analysed phase precession place and grid cells with clear in theta-modulation – showing normal phase precession in VR as in R (with a lower slope consistent with the larger firing fields in VR, but the correlation between slope and field size only reaching significance for place fields in R). These results indicate that theta phase precession in place and grid cells is independent of linear vestibular acceleration signals and the absolute value of theta frequency. Text has been added to the last paragraph of the Results subsection “Electrophysiology”, to the eighth paragraph of the Discussion, and in the Abstract. Methods are included in the last paragraph of the subsection “Firing rate map construction and spatial cell classification”.